# Soil Reinforcement with Geocells and Vegetation for Ecological Mitigation of Shallow Slope Failure

**Gaofeng Song [1],\*, Xiaoruan Song [1], Shiqin He [1], Dezhong Kong [2] and Shuai Zhang [3]**

1   School of Civil Engineering, North China University of Technology, Beijing 100144, China
2   Mining College, Guizhou University, Guiyang 550025, China
3   School of Architecture and Design, Beijing Jiaotong University, Beijing 100044, China
\*   Correspondence: song.gaofeng@ncut.edu.cn

**Abstract:** Soil reinforcement using geocells and vegetation is one of the best forms of soil protection for shallow slope failure control. The geocell supports the vegetation growth and the vegetation cover provides protection against the surface erosion. This work proposed a soil treatment method using geocells for supporting the vegetation growth and stabilizing the shallow slope. A step-by-step installation of the geocells in the field and the development of vegetation growth were also described. The authors developed nine physical models that were reinforced with different sized geocell structures (no reinforcement and small and large geocell reinforcement). The models were placed under three rainfall intensities (50, 75, and 100 mm/h). The stability of the slope under the rainfall and the performance of the geocell reinforcement were assessed from the the development of slope failures, the soil erosion and the slope displacement. The results showed that the stability of geocell reinforced slopes were better off than the unsupported slope. The small geocell-reinforced slopes showed less measured soil erosion and also smaller slope displacement. In general, small geocells outperformed large geocells in terms of the erosion control and slope stabilization. The rainfall intensity dramatically increased the soil erosion on slopes. The geocell- and vegetation-treated slope in the field showed good resistance against the surface erosion.

**Keywords:** soil reinforcement; slope stabilization; erosion control; vegetation growth; geocell

## 1. Introduction

Shallow slope failure is the superficial removal of the small-scale topsoil and vegetation cover from the slopes [1,2], which can be classified into the shallow landslide and the shallow erosion [3]. Shallow landslides may occur when the forces acting on the downslope exceed the mechanical resistance of the slope soil [4–6]. Shallow soil erosion, on the other hand, mostly occurs under a heavy rainfall as the rainwater dislodges soil particles and carries them off a lope [7]. Shallow slope failure is commonly observed in areas of hilly or mountainous terrain, engineered roads and embankments [8,9]. Shallow slope failure is a worldwide serious geologic hazard and causes severe problems in both the natural environment and human properties [10–14].

Soil erosion treatment with vegetation is one of the most effective methods for controlling the surface erosion and improving the slope stability [15–17]. The new vegetation cover on slopes provides protection against the surface erosion both mechanically and hydrologically [2,18,19]. The vegetation root structures penetrated into the topsoil provide strength to the soil by holding it in place to resist movement. The cover of the vegetation on the top also protects the slope against the wind and rain. Other benefits such as slowing down the water velocity, removing moisture from the soil, and maintaining soil porosity are also identified. Properly designed and planted vegetation on steep slopes plays an important role in preventing the long-term erosion and the subsequent movement of the shallow slope mass [20–22]. Most of the vegetative measures are also inexpensive, cover a

large area on the slope, and look harmonious with the natural landscape for the esthetic consideration [23].

As one of the best forms of soil protection, vegetation often requires a stronger engineered product such as the geocell to support it. This is because in some cases, when not supported, the vegetation cover on slopes can slough off on mass before it starts to stabilize the soil [24,25]. The stability of the new vegetation in the early stage should be ensured. In areas unlikely to establish natural vegetation, engineered erosion products (geocells) may be used to support vegetation growth and protect embankments and infrastructure. For more challenging conditions such as extremely steep slopes and high runoff flow, more effective and robust products are necessary. The product should be fixed securely and provide protection for vegetation. A geocell can be used to hold slope soil within its profile, maintain the integrity of the soil surface, and support vegetation growth. A number of advantages are identified [26–28]. The flat and collapsible geocell allows easy and low-cost transportation. The cost-effective structure provides rapid installation with the use of the local available material. The installation only requires minimal use of machinery, low manpower, and unskilled labor to allow the vegetation growth on slopes.

In this work, a composite soil reinforcement method using the geocell structures and the wheat straws was proposed for vegetation support and slope stabilization [29,30]. The geocell structures were first placed on the prepared bare slope; the local soil was utilized to fill the geocell sections. A thin layer of the soil–straw mixture was placed on top of the treated slope to retard the water runoff. They may also increase the biological activities in the soil, modify the level of available nutrients, and help to maintain or increase the level of soil organic matter [21,22,31,32]. Seeding was then performed after the slope treatment to establish vegetation. This composite reinforcement method modified the soil surface and topography of the bare slope to control the runoff and support vegetation growth. This work attempted to study the soil reinforcement and slope stabilization using geocells and wheat straws from physical models and field practice. The work had three main goals: (1) to develop a number of physical models with different sized geocell reinforcement and rainfall intensities; (2) obtain the progressive development of slope failures by assessing the soil erosion characteristics, the amount of soil erosion, and the slope displacement; (3) perform the soil treatment and vegetation growth in the field for slope stabilization.

## 2. Physical Modeling Study

### 2.1. Model Development

2.1.1. Model Description

The authors designed nine physical models in this research (Table 1). The models were set up at a slope ratio of 1:1.5 (rise/run). The three models in Group A were constructed using the bare soil with no extra reinforcement. The slopes in Group B were reinforced using the wheat straw and small geocells. The small geocell was 370 mm long, 370 mm wide, and 150 mm high. In Group C, the large geocells (445 × 445 × 200 mm) were utilized. The models were placed under three different rainfall intensities at 50 mm/h, 75 mm/h, and 100 mm/h, which were determined from the rainfall condition at the studied site in Yinchuan city.

**Table 1.** Soil reinforcement and rainfall intensity for different models.

| Models | Soil Reinforcement | Geocell Size | Rainfall Intensity (mm/h) |
|---|---|---|---|
| $A_1$ | | | 50 |
| $A_2$ | Bare soil | - | 75 |
| $A_3$ | | | 100 |
| $B_1$ | | | 50 |
| $B_2$ | Geocell and wheat straw reinforcement | Small | 75 |
| $B_3$ | | | 100 |
| $C_1$ | | | 50 |
| $C_2$ | Geocell and wheat straw reinforcement | Large | 75 |
| $C_3$ | | | 100 |

### 2.1.2. Modeling Rig and Model Construction

A physical modeling rig with the dimensions of 3000 mm long, 1200 mm wide, and 500 mm high was used in this research to investigate the performance of the soil slopes under the impact of rainfall. A thin steel plate was forged in the middle of the rig to include two physical models. Local soil was used to infill the modeling rig for model construction. Wheat straws from the current season with good resilience were collected and mixed with the soil before infilling. The diameter of the straws is about 5 mm. These straws were cut to 3–5 mm long and fully mixed with the collected soil. To ensure the best reinforcement, the ratio of straw reinforcement (the ratio between the weight of wheat straws and the weight of the soil–straw mixture) was determined as 0.5%.

The physical models in Group A contained the bare soil with no extra reinforcement. For the geocell and straw reinforced slopes, the geocells were placed on the compacted soil layer at the bottom, and were infilled with the mixture of soil and straw. The infilled soil was properly compacted until the designed height was reached. The rear side of the rig can be lifted to adjust the slope steepness. The finished model is shown in Figure 1.

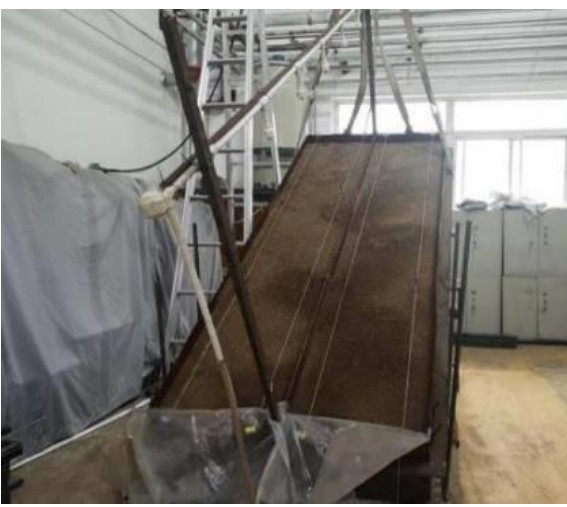

**Figure 1.** The constructed physical model at a slope ratio of 1:1.5.

The scaling law is important to the physical study for obtaining accurate and meaningful results. The similarity principles were followed when performing the physical modeling test. Since the full-size geocells were used in the experiment, the geometry coefficient was selected as 1. The density and strength similarity coefficients were also determined as 1. The soils were compacted properly when constructing the physical models so that the density and strength of the modeling slopes were close to the real case.

Five sprayers were fixed evenly above the modeling rig to simulate the rainfall [33,34]. The whole area of the slopes was covered by the artificial rainwater. The average annual precipitation in Guyuan is around 500–600 mm. Intense individual downpours for a relatively short time are not unusual in July and August. Sometimes, the rainfall intensity of the individual downpours can be extremely high. The measured heaviest rainfalls to occur in 24 h and 1 h were more than 110 mm and 50 mm, respectively. According to the rainfall condition on the studied site, the three rainfall intensities in this experiment were selected as 50, 75, and 100 mm/h. Each rainfall continued for 30 min. The high levels of rainfall intensities were selected in this study for considering the worst-case scenario. The rainwater was collected and measured from the runoff ports forged at the front edge of the rig. The amount of soil erosion and the slope displacement were also measured in this work. The amount of soil erosion was determined from the collected water and soil at the runoff ports. The displacement of the slope at the top section was measured using the ruler fixed on the modeling rig.

### 2.2. Experimental Results

#### 2.2.1. Slope Failures

Figure 2 shows the failure characteristics of the bare slopes under different rainfall intensities after 30 min of rainfall. A number of cracks were observed on the bare slope in Model $A_1$ at the rainfall intensity of 50 mm/h (Figure 2a). However, the inverse impact of the raindrop to the slope was not significant, and the slope surface still remained smooth and flat. By contrast, an increase in the rainfall intensity contributed to the soil degradation in Models $A_2$ and $A_3$. Under the rainfall intensity of 75 mm/h, the soil erosion was insignificant at the top section of the slope, but the soil deterioration was clearly observed at the bottom (Figure 2b). As the rainfall intensity increased to 100 mm/h, rill erosion was found over the majority part of the slope surface (Figure 2c).

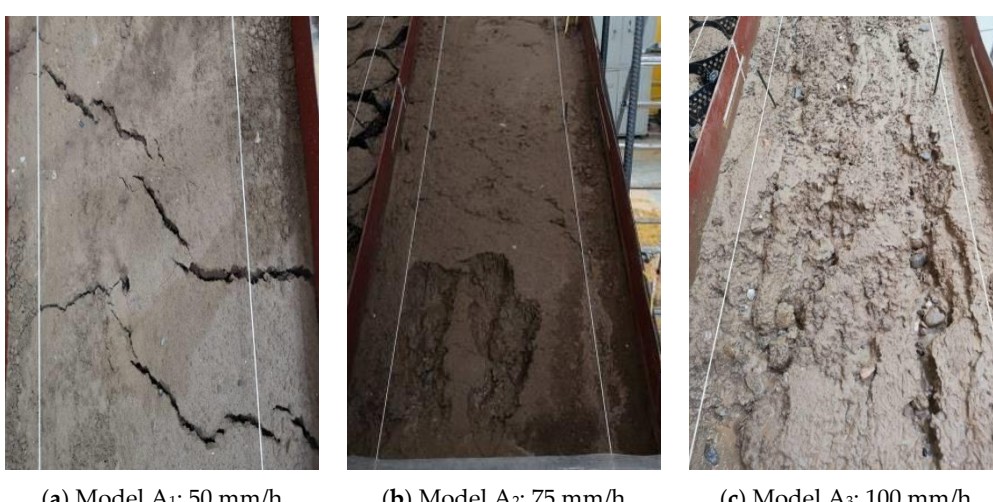

(**a**) Model $A_1$: 50 mm/h      (**b**) Model $A_2$: 75 mm/h      (**c**) Model $A_3$: 100 mm/h

**Figure 2.** Failure characteristics of the bare slopes after the rainfall.

The stability and surface erosion of the geocell reinforced slopes were better off as compared with the bare slope. Figure 3 shows the final appearance of the small geocell-reinforced slopes under the three rainfall intensities. The overall integrity of the geocell-reinforced slopes was maintained. No cracks or any gully and rill erosion were observed on the slope surfaces reinforced by geocells. The geocell structures played an important role in improving the slope stability by holding the soil in place. On the other hand, the soil degradation increased with the rainfall intensity, which was observed from the depth of the re-appeared geocell section.

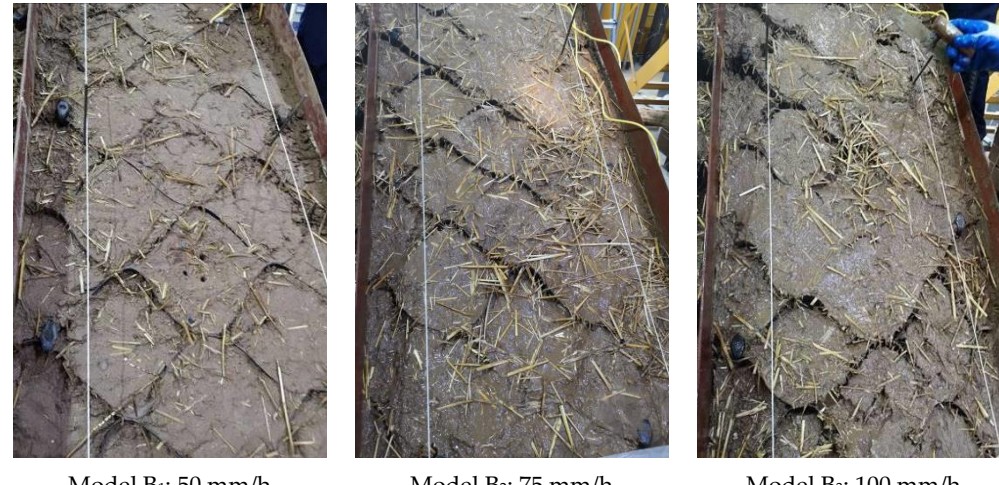

| Model B₁: 50 mm/h | Model B₂: 75 mm/h | Model B₃: 100 mm/h |

**Figure 3.** Failure characteristics of the small geocell-reinforced slopes after the rainfall.

The large geocell-reinforced slopes after 30 min of rainfall are shown in Figure 4. Again, the rainfall reduced the slope stability significantly, and an increase in the rainfall intensity resulted in more soil deterioration. The soil erosion and the stability of slopes were also compared to those reinforced with small geocells in Figure 3. It was noted that, in Figure 4, a larger depth of the re-appeared geocell sections was observed, indicating an increased amount of soil erosion. Hence, the small geocells outperformed the large geocells in terms of the soil erosion control.

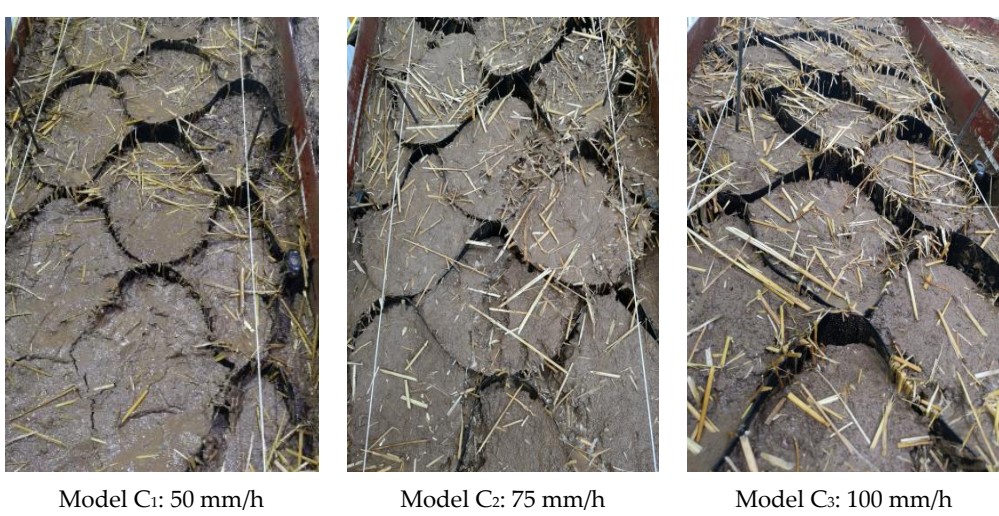

| Model C₁: 50 mm/h | Model C₂: 75 mm/h | Model C₃: 100 mm/h |

**Figure 4.** Failure characteristics of the large geocell-reinforced slopes after the rainfall.

2.2.2. Soil Erosion

Figure 5 shows the growth of soil erosion with the rainfall process for different slopes. The amount of soil erosion was almost unnoticeable under the rainfall intensity of 50 mm/h. However, the soil erosion increased with the rainfall intensity at an even faster rate. The bare slope saw 550 g of soil erosion at the end of the 50 mm/h rainfall, which was significantly raised to about 9000 g and 22,000 g at the rainfall intensities of 75 mm/h and 100 mm/h, respectively (Figure 5a). The soil erosion was decreased when the slope was reinforced with the small geocells and the wheat straw. The amounts of soil erosion for the three slopes were observed as 40 g, 1400 g, and 3900 g at the cease of the rainfall. The large geocells also decreased the soil erosion; the amounts of soil erosion were 30 g, 1500 g and 5000 g under the three rainfall intensities. However, the small geocell-reinforced models showed less soil erosion; therefore, the small geocells outperformed the large ones.

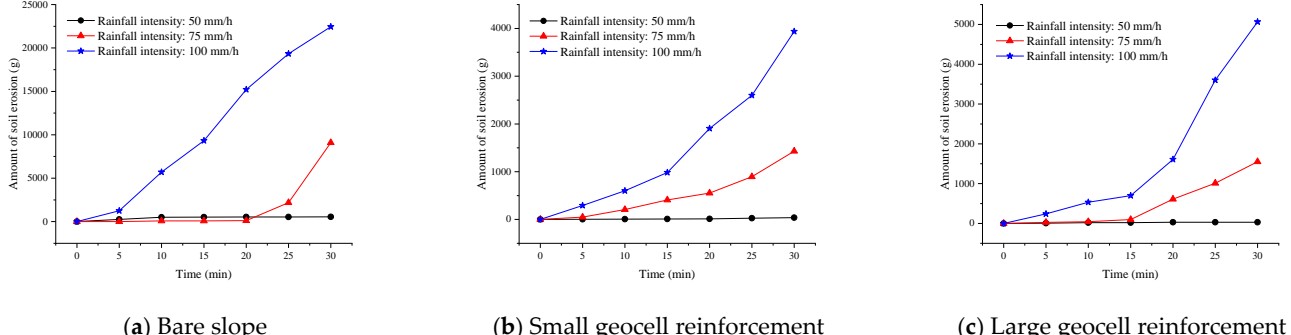

**Figure 5.** Amount of soil erosion for different reinforced slopes under the three rainfall intensities.

### 2.2.3. Slope Displacement

The displacement at the top section of the slope was recorded during the test until 30 min after the rainfall (see Figure 6). The slope displacement increased at the first 30 min as the rainfall continued. The increase in slope displacement slowed down and started to stabilize after the rainfall ceased. The increase in the rainfall intensity increased the slope displacement. For the bare slope, the final slope displacement grew from 4 mm to 7.6 mm and 12 mm as the rainfall intensity increased (Figure 6a). The geocells improved the slope stability with decreased slope displacement. Particularly, the slope displacement was better controlled with the small geocells. This was seen from the fact that, after the rainfall ceased for 30 min, the small geocell-reinforced slopes found the displacement stabilizing at 1 mm, 2 mm, and 2.6 mm under the three rainfall intensities, respectively (Figure 6b). By comparison, the slope displacement with the large geocells stood at 2 mm, 4.4 mm, and 5.3 mm (Figure 6c).

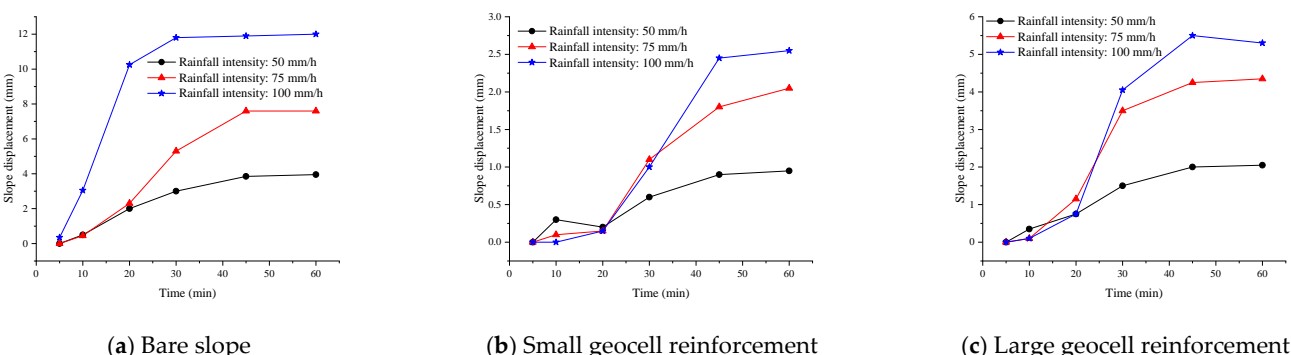

**Figure 6.** Amount of soil erosion for different reinforced slopes under the three rainfall intensities.

## 3. Filed Practice

### 3.1. Geocell Installation

#### 3.1.1. Site Preparation

The first step for the slope stabilization with geocells and vegetation was the site preparation. An area of 10 m × 10 m was first selected on the subgrade at the studied area (Figure 7a). The selected area was then properly prepared by removing all the debris, rocks, sticks, weeds, vegetation and gullies before installing the geocell structures (Figure 7b). Voids or channels were filled so that uncontrolled water flow was not allowed under the geocells. The surface was properly leveled and compacted to ensure a smooth and fine graded slope surface. The finished slope after preparation is shown in Figure 7b.

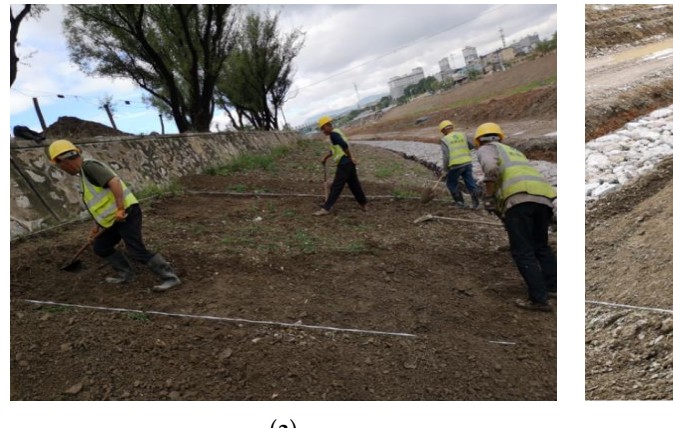
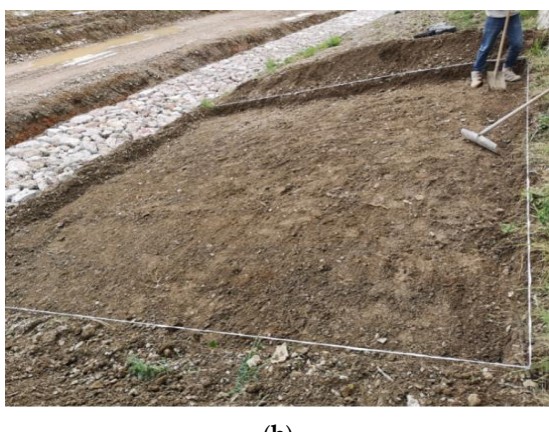

| (**a**) | (**b**) |

**Figure 7.** Site preparation: (**a**) slope preparation; (**b**) the slope after preparation.

### 3.1.2. Geocell Installation and Infilling

The next step was the installation of the geocells over the slope and infilling. The geocell was held in place at the crest of the slope with anchors, and fully expanded down the slope and secured at the toe of the slope (Figure 8a). The geocells were opened to a fixed width using separator bars and fixed to the ground using U-shaped nails at a proper interval. A full contact between the cell and the slope surface was properly maintained. After the geocell sections were secured on the prepared surface, infills were placed into the cells. The infills for this study were the local topsoil with appropriate nutrients. The bottom 10 cm of infill placed into the cells was the sufficient bare soil extracted from the neighbouring site (Figure 8a), while the top 10 cm infill was the mixture of the bare soil and wheat straws. Infilling was performed from the crest to the toe (Figure 8b). The topsoil was spread uniformly and evenly over the entire surface. The soil materials were overfilled 25 mm and compacted to the designed height. Excess soil was scraped off from the surface. The treated slope surface is shown in Figure 8c.

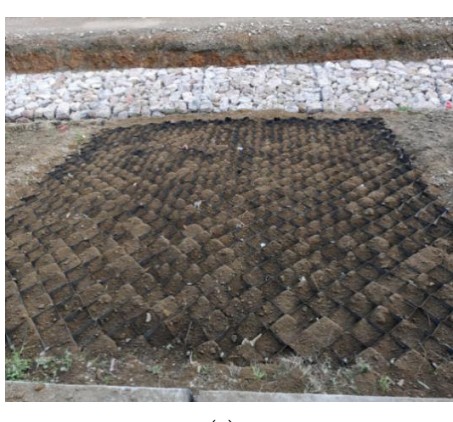
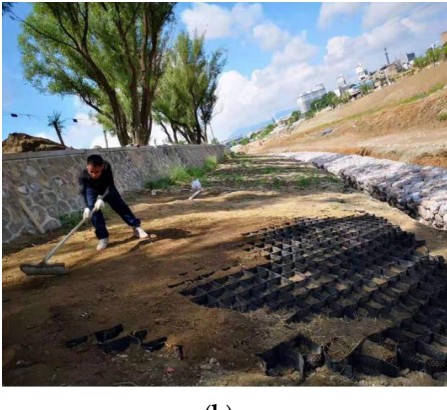
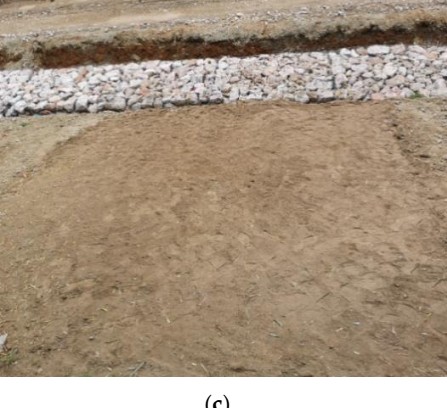

| (**a**) | (**b**) | (**c**) |

**Figure 8.** Infilling of the top soil: (**a**) bottom 10 cm of bare soil; (**b**) infilling from the crest to the toe; (**c**) the finished slope after treatment.

### 3.1.3. Seeding

The third step in the ecological restoration process of the slope was to seed the studied area to be vegetated. The seed is suitable to the soil condition at the studied site. It can also be easily adapted to the local climate and geographical area. The vegetation also has a fast growth rate. The fertilizer is rich in nitrogen, phosphorus, and potassium. The fertilizer was mixed with the topsoil before infilling. In this work, the seed was placed by hand

over the entire surface on 18 July 2019. Once infilled with all the soil materials, an erosion control blanket was used to cover the treated slope face (Figure 9).

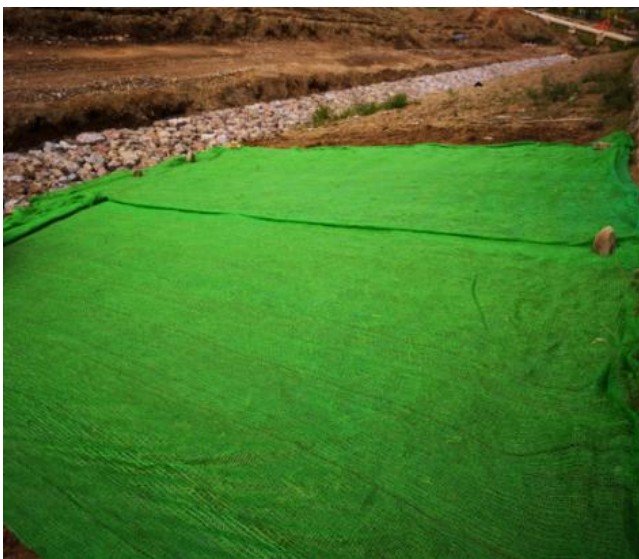

**Figure 9.** Slope surface covered with blanket after seeding.

### 3.2. Development of Vegetation

The vegetation growth 13 days after seeding is shown in Figure 10. The fast-growing vegetation was well developed beneath the erosion control blanket on the geocell- and wheat straw-treated slope (Figure 10a). The slope was covered with the vegetation over the entire slope surface within the selected area. This was compared to the neighboring untreated bare slope with only few plants on the sandy ground (Figure 10b). As compared with the treated slope soil in the studied area, the sandy soil on the bare slope was unable to provide sufficient and proper nutrients for the vegetation growth. The poor water conditions on the bare slope were also unfavorable to the growth of new vegetation. Vegetation can hardly be planted on the bare slope unless properly treated.

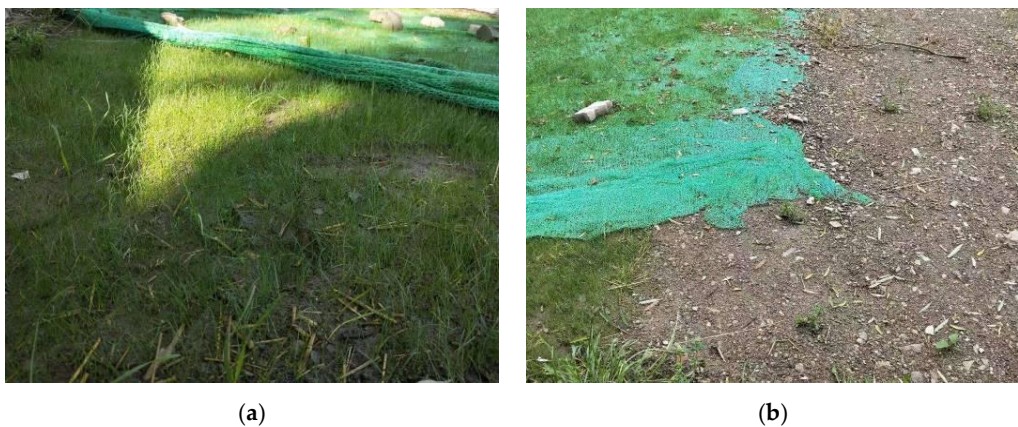

(**a**)　　　　　　　　　　　　　　　　(**b**)

**Figure 10.** Vegetation growth at the 13rd day after seeding: (**a**) vegetation growth within the selected area; (**b**) comparison of the vegetation growth on the treated and bare slopes.

A rainstorm event occurred at the studied site on 4 August 2019, the 17th day after seeding. The slope condition after the impact of the rainstorm is shown in Figure 11. The vegetation was observed on most of the area above the erosion control blanket (Figure 11a). The treated slope in the selected area was neat and clean, with only limited rock and wood debris found at the edge of the selected site, which might be shifted from the neighboring

untreated area by the rainwater. This indicates that the slope in the area covered with the erosion control blanket was barely influenced by the rainstorm. By contrast, no protection against erosion was provided to the neighboring untreated slope. As a result, the bare slope saw a large amount of rock debris scoured around by the rainwater (Figure 11).

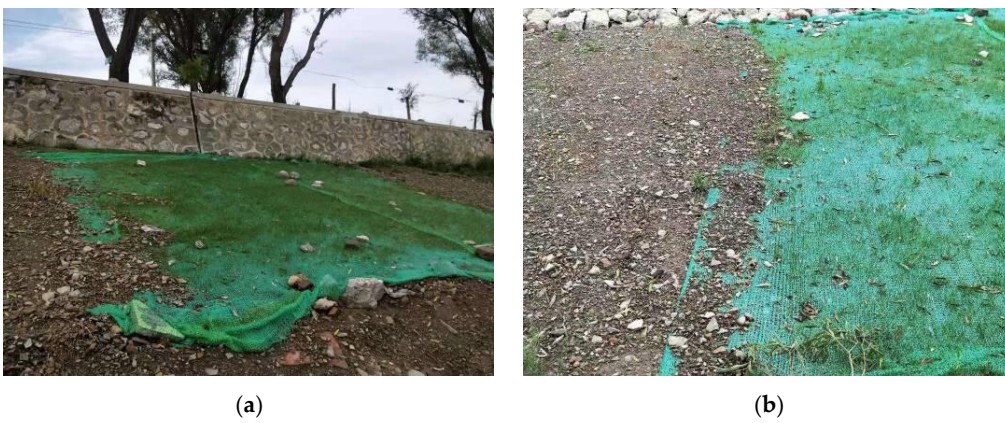

(**a**)          (**b**)

**Figure 11.** The condition of the slope under the impact of the rainfall on the 17th day after seeding. (**a**) vegetation observed above the blanket; (**b**) comparison of conditions of the treated and bare slopes.

The establishment of the vegetation after 75 days and 120 days of seeding is shown in Figure 12a,b respectively. A picture of the vegetation was taken from the other side of the river on 1 October 2019 (Figure 12a). It was observed that that the vegetation was well developed on the selected slope surface, in strike contrast to the neighboring untreated slope area with only few plants observed. The new vegetation provided protection against the surface erosion. Four months after seeding, another visit was paid to the studied site on 15 November 2019. The grass in the studied area turned brown in the winter time (Figure 12b). As compared with the bare slope, however, the vegetation cover still protected the slope against the wind, rain and snow. The root system of the vegetation and the geocell structures also provided strength to the soil by holding it in place.

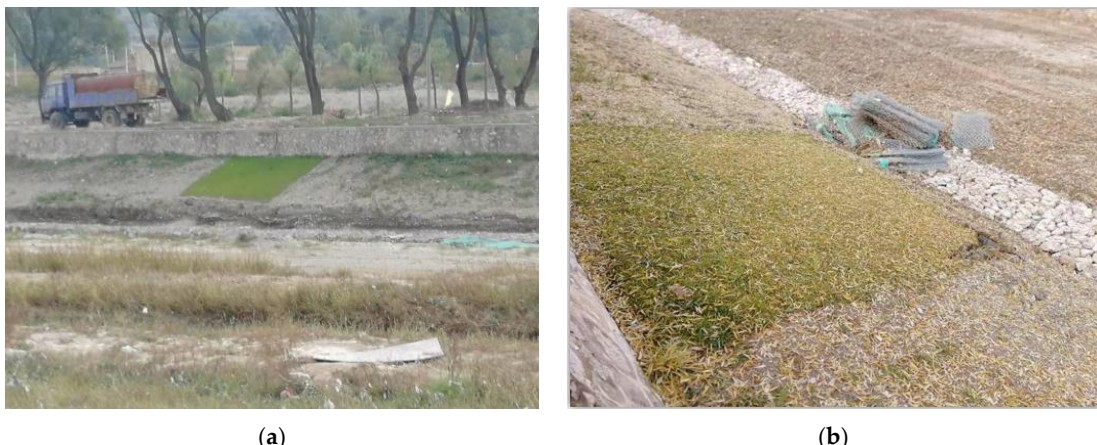

(**a**)          (**b**)

**Figure 12.** Establishment of vegetation after (**a**) 75 days and (**b**) 120 days.

### 4. Summary and Conclusions

This work proposed a vegetative and soil treatment measure for soil erosion control and slope stabilization. The geocells were used to support the vegetation growth. The vegetation protected the slope against the surface erosion and provided strong root strength to resist movement. The filled soil–straw mixture enhanced nutrients in soil after decomposition. A total of nine physical slopes were prepared in this work; the models were reinforced with different-sized geocells and were placed under different rainfall intensities.

The development of slope failures, soil erosion, and surface displacement were recorded and analyzed. The geocell installation and seeding were also performed at the selected area in the field. Important findings are listed below.

The slope stability was largely improved by the geocell and wheat straw reinforcement. Horizontal cracks and rill erosion were clearly observed on the bare slopes. By contrast, the reinforced models maintained a good integrity and overall stability. The stability improvement by geocells and wheat straws were also verified from the soil erosion and slope displacement. The amount of soil erosion for the bare slope reached 22,000 g under a heavy rainfall, which was compared with below 5000 g for the reinforced slopes. Similarly, the largest displacement of the bare slope was observed at 12 mm, compared with 3–5 mm for the reinforced models.

The small geocells outperformed the large geocells in terms of the soil erosion control. More loss of soil was clearly seen from the large geocell-reinforced slopes, which was also validated from the measured amount of soil erosion with 3900 g of soil loss for the small geocell-reinforced slopes and 5000 g for the large geocell-reinforced slopes. The slope displacement further confirmed this trend, as the largest slope displacement for the large geocell slopes was 5.3 mm, compared with only 2.6 mm for the slopes reinforced with small geocells.

The rainfall had a significant adverse influence on the slope stability. The soil condition and slope stability deteriorated with the rainfall process. On the other hand, the amount of soil erosion and the slope displacement were dramatically increased with an increase in the rainfall intensity. The soil erosion was unnoticeable at the low rainfall intensity. It rose rapidly with the rainfall intensity. Likewise, the slope displacement tripled as the rainfall intensity increased from 50 to 100 mm/h.

The installation of the geocells in the field included the site preparation, geocell installation, infilling, and seeding. The geocells were secured at the crest and toe of the slope, and were expanded down over the slope surface. U-shaped nails were used to fix the geocells to the slope at a proper interval, so that an intimate contact with the soil was ensured. The treated slope face was covered with an erosion control blanket after seeding. The vegetation was well developed after 13 days of seeding. Compared with the untreated slope, the geocell-reinforced and vegetation-treated slope in the studied area showed good resistance to the impacts of the rainstorm on the 17th day after seeding. The fully-developed vegetation started to provide protection to the slope against wind and rainfall; the root system worked as integrated with the geocell structures to provide mechanical reinforcement to the slope.

**Author Contributions:** All authors contributed to the study conception and design. Material preparation, data collection and analysis were performed by G.S., X.S., S.H., D.K. and S.Z. The first draft of the manuscript was written by G.S. and all authors commented on previous versions of the manuscript. All authors have read and agreed to the published version of the manuscript.

**Funding:** The paper was supported by National Natural Science Foundation of China (52004010), R&D Program of Beijing Municipal Education Commission (KM202010009001) and the Excellent Research Team of North China University of Technology (107051360021XN083/019).

**Institutional Review Board Statement:** Not applicable.

**Informed Consent Statement:** Not applicable.

**Data Availability Statement:** Data is contained within the article.

**Conflicts of Interest:** The authors have no relevant financial or non-financial interests to disclose.

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
