# Peer review of "Soil Reinforcement with Geocells and Vegetation for Ecological Mitigation of Shallow Slope Failure"

_sustainability, doi:10.3390/su141911911_

Round 1
Reviewer 1 Report
Using geocells and vegetation for soil reinforcement is an environmentally-friendly method for shallow slope stabilization. The manuscript provides an interesting physical modeling test to investigate the reinforcement of two different sized geocells (plus the bare slope for comparison) under three different rainfall intensities. The authors also provide a detailed description on the field application of the geocell and vegetation reinforced method on shallow slopes, including the site preparation, geocell installation and infilling, seeding. The manuscript should provide implication for future study. The manuscript is acceptable but minor revisions are requested before acceptance.
1. The abstract should be overhauled. Authors should provide quantitative results and “take-home” messages in the abstract. I don’t see “take-home” messages from the abstract.
2. The manuscript focuses on the soil erosion control on the shallow slope using the geocells and vegetation. “Shallow slope failure” is also seen in the manuscript title, but the authors did not give a description on shallow slope failure in Introduction. The reviewer recommends the authors to provide a brief introduction on the shallow slope failure at the beginning of the manuscript, say before introducing the soil erosion treatment with vegetation.
3. The simulated rainfall intensity in this research (100 mm/h) is extremely high. Please justify your selection of rainfall intensity.
4. Some of the pictures are not necessary. For example, Figure 8b, Figures 9a and 9b are completely unnecessary. Please consider deleting the unnecessary pictures from the manuscript.
5. The preparation of the physical modeling test and the field practice are well described.
6. The bare slope is inversely impacted by the rainstorm. Authors use “…… is inversely/positively/negatively impacted by ……” in the manuscript to report the experimental result, but by what extent? That is not a scientific expression for a scientific paper.
Reviewer 2 Report
The manuscript is generally well organized. The methods are well explained and the analysis is adequate. The results are reasonable. It should provide guidance or reference for future similar research in this field. After reviewing the manuscript, the following recommendations should be considered by the authors.
1. “paper” to “study” or “work”.
2. The “engineered product” used in this research is the “geocell” instead of other engineered products. I would recommend using “geocell” rather than “engineered product” in the introduction.
3. Please use the “past tense” to report what happened in the past: what someone reported? What the authors did? What happened in an experiment?
4. Paragraph 3, Introduction: what is a “geocell room”?
5. Both “rainwater” and “rain water” are found in the manuscript. Please, use “rainwater”.
6. Summary and conclusion: The filled soil-straw mixture enhances the mechanical properties of the reinforced soil. Is this a correct statement? How is this conclusion reached from the manuscript?
7. Summary and conclusion: confirm not conform. The manuscript has some grammar and language issues. Authors need to address the language errors and further proofread the manuscript.

Reviewer 3 Report
1. The title of the paper is too broad and lacks pertinence. It does not highlight the specific technical means of the application of the paper,such as Geocell and wheat straw reinforcement.
2. In the physical experiment part of the paper, the effect of doping wheat straw is not explained clearly, and the influence of doping proportion on soil reinforcement is not analyzed.
3. In the field practice part, there are only photos of the construction process, but there are no slope reinforcement results after the implementation, which does not correspond well with the physical model in the first half of the paper and the theme of the paper.
Author Response
1. The reviewer is correct. Authors have changed the title to "Soil reinforcement with geocells and vegetation for ecological mitigation of shallow slope failure".
2. Authors have provided the explanation on the effect of doping wheat straw and the doping proportion. Please see the revised manuscript and also below.
Wheat straws from the current season with good resilience were collected and mixed with the soil before infilling. The straws secured to the slope models can retard the water runoff and also increase the biological activities in the soil with improved level of available nutrients. The improved fertility in soil is beneficial to the vegetation establishment on the slope. The diameter of the collected straws is about 5 mm. These straws were cut to 3–5mm long and fully mixed with the collected soil. To ensure the best reinforcement, the ratio of straw reinforcement (the ratio between the weight of wheat straws and the weight of the soil-straw mixture) was determined as 0.5%.
3. Authors were expected to observe the development of vegetation on the treated slope in the studied site and analyze its protection against any slope failure. Unfortunately, no further visit was paid to the studied site because of the coronavirus pandemic. However, compared to the neighboring untreated bare slope in the field, it was seen that the treated area showed good development of the vegetation and provided protection against the rain and wind (see Development of vegetation in the manuscript).
Round 2
Reviewer 3 Report
no more comments.